# 3D visualization of mitochondrial solid-phase calcium stores in whole cells

**Sharon Grayer Wolf[1]\*, Yael Mutsafi[2], Tali Dadosh[1], Tal Ilani[2], Zipora Lansky[2], Ben Horowitz[2], Sarah Rubin[3], Michael Elbaum[3†], Deborah Fass[2]\***

[1]Department of Chemical Research Support, Weizmann Institute of Science, Rehovot, Israel; [2]Department of Structural Biology, Weizmann Institute of Science, Rehovot, Israel; [3]Department of Materials and Interfaces, Weizmann Institute of Science, Rehovot, Israel

**Abstract** The entry of calcium into mitochondria is central to metabolism, inter-organelle communication, and cell life/death decisions. Long-sought transporters involved in mitochondrial calcium influx and efflux have recently been identified. To obtain a unified picture of mitochondrial calcium utilization, a parallel advance in understanding the forms and quantities of mitochondrial calcium stores is needed. We present here the direct 3D visualization of mitochondrial calcium in intact mammalian cells using cryo-scanning transmission electron tomography (CSTET). Amorphous solid granules containing calcium and phosphorus were pervasive in the mitochondrial matrices of a variety of mammalian cell types. Analysis based on quantitative electron scattering revealed that these repositories are equivalent to molar concentrations of dissolved ions. These results demonstrate conclusively that calcium buffering in the mitochondrial matrix in live cells occurs by phase separation, and that solid-phase stores provide a major ion reservoir that can be mobilized for bioenergetics and signaling.

DOI: https://doi.org/10.7554/eLife.29929.001

**\*For correspondence:**

sharon.wolf@weizmann.ac.il (SGW);

deborah.fass@weizmann.ac.il (DF)

**Present address:** †Department of Chemical and Biological Physics, Weizmann Institute of Science, Rehovot, Israel

**Competing interests:** The authors declare that no competing interests exist.

## Introduction

Cells use ion gradients for information transfer and energy storage. A widely known example is the proton gradient in mitochondria, which allows for oxidative phosphorylation in the production of ATP (*Saraste, 1999*). Calcium gradients have an equally prominent role in cell physiology, and mitochondria are central players in calcium storage and utilization (*Szabadkai and Duchen, 2008*). Calcium ions affect mitochondrial physiology by regulating respiratory chain complexes, tricarboxylic acid cycle proteins, and metabolite transporters (*Glancy and Balaban, 2012*). Furthermore, substantial potential energy is stored in the calcium gradient across the inner mitochondrial membrane (*Glancy and Balaban, 2012*). Calcium uptake by mitochondria also influences cellular calcium signaling (*Rizzuto et al., 2012*) and, in excess, can trigger apoptotic cell death (*Giorgi et al., 2012*). For all these reasons, a thorough understanding of how calcium is handled by mitochondria is immensely important.

Despite the progress in identifying and characterizing the molecules responsible for mitochondrial calcium uptake and release (*Jiang et al., 2009*; *Palty et al., 2010*; *De Stefani et al., 2011*; *Baughman et al., 2011*), an appreciation for the quantities, forms, and availability of calcium in mitochondria has been hampered by inadequate preservation of ions with typical imaging and detection methods. Although fluorescent indicators are widely used in the study of mitochondrial calcium dynamics (*Pendin et al., 2015*), these sensors inevitably perturb the soluble ion pool. Moreover, they are blind to solid-phase stores or tightly bound species. Transmission electron microscopy (TEM) of metal-stained thin cell sections has for decades provided details on mitochondrial

morphology and interactions (*Palade, 1953*; *Lee et al., 2016*), but ions and other unfixed contents are likely to be flushed from the organelle during sample preparation.

Scanning transmission electron microscopy (STEM) is a technique that allows for mapping of samples with sensitivity to atomic number and has a long history of applications in materials science (*Pennycook, 2012*). STEM tomography of cryo-preserved samples is currently being developed for its unique applications to biological systems (*Wolf et al., 2014*). Following a study of polyphosphate bodies in bacteria (*Wolf et al., 2015*), we used cryo-STEM tomography (CSTET) to examine ion deposition in mammalian cells. We report that CSTET provides a faithful view of ion storage in mitochondria, revealing the quantities and organization of accumulated calcium in natural intracellular contexts.

## Results and discussion

### Cryo-STEM tomography reveals morphology and interactions of whole mitochondria within cells

For CSTET visualization, cells were grown in standard culture conditions and prepared for electron microscopy by vitrification. Unless otherwise noted, cells were vitrified 2 or 3 days after seeding onto microscopy grids, when they were well-adherent and showed evidence of proliferation but

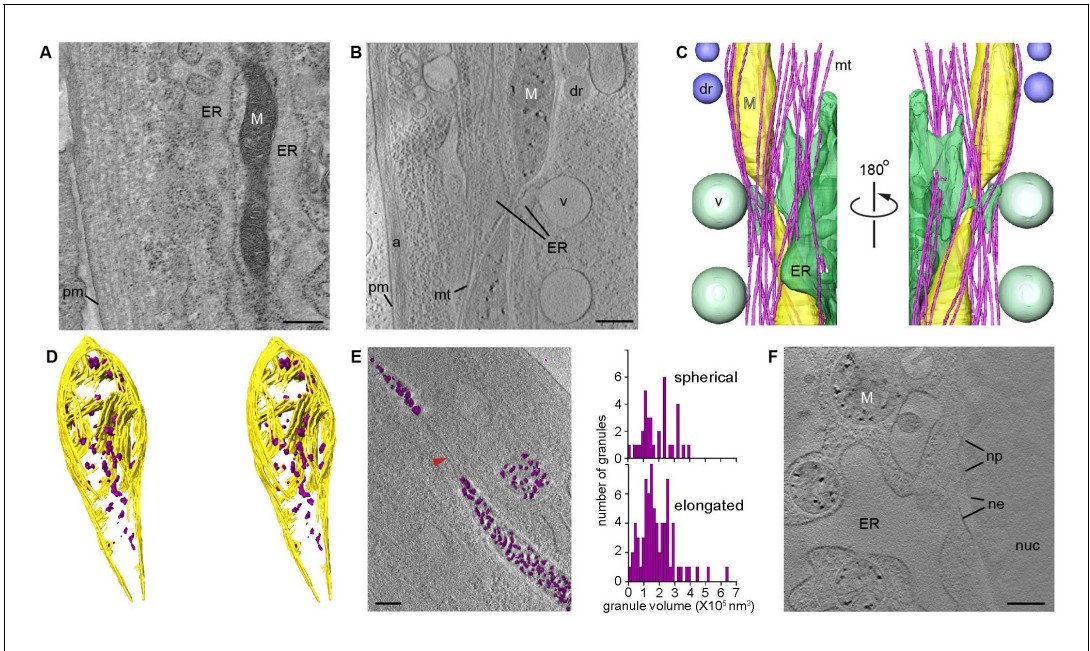

**Figure 1.** CSTET imaging shows whole mitochondria in situ in mammalian cells. White M indicates mitochondria; pm, plasma membrane; ER, endoplasmic reticulum; mt, microtubules; dr, lipid droplet; v, vesicle; a, actin; nuc, nucleus; ne, nuclear envelope; np, nuclear pore. Scale bars are 400 nm. (**A**) Conventional TEM image of a heavy metal stained thin section of a human embryonic lung (WI-38) fibroblast shows staining artifacts, particularly of the mitochondrion. (**B**) A 30-nm thick section of a CSTET reconstruction ( 750 nm thickness in total) of a WI-38 fibroblast. (**C**) Segmentation of the CSTET reconstruction. (**D**) Stereo pair of segmentation of the upper mitochondrion from panels B and C revealing internal ultrastructure. Mitochondrial membranes and cristae are yellow, granules are purple. (**E**) Elongated and spherical mitochondria show similar granule size distributions. Granule volumes were segmented using an intensity threshold and are displayed in purple above a section from the corresponding tomogram. Red arrowhead indicates a fission tubule. (**F**) Mitochondria near the nucleus of a human dermal microvasculature endothelial cell (30-nm thick section from a region of 790 nm total thickness).

DOI: https://doi.org/10.7554/eLife.29929.002

The following figure supplements are available for figure 1:

**Figure supplement 1.** CSTET visualization of mitochondrial interactions.
DOI: https://doi.org/10.7554/eLife.29929.003
**Figure supplement 2.** Matrix granules are ubiquitous in mitochondria.
DOI: https://doi.org/10.7554/eLife.29929.004

were still sub-confluent. Tomographic data were collected without further manipulation, preserving native interactions among organelles (*Murley and Nunnari, 2016*) and the physiological distribution of chemical constituents. We note that CSTET is well suited to provide deep views (up to >1 μm) into cells (*Rez et al., 2016*). For comparison, conventional cryo-TEM tomography is restricted to thin cell regions (preferably less than 300 nm) (*Villa et al., 2013*) or requires technically challenging processing of thicker regions by cryo-microtomy (*Ladinsky et al., 2006*) or focused-ion beam milling (*Marko et al., 2007*; *Mahamid et al., 2016*). Conventional metal-stained thin sections of fixed cells (*Figure 1A*) are approximately 70-nm thick and unavoidably display staining artifacts. The value of imaging thick cell regions in unperturbed cells is emphasized by a CSTET reconstruction and segmentation from a region of a WI-38 fibroblast cell with a depth of 750 nm (*Figure 1A,B,C* and *Videos 1–6*). Endoplasmic reticulum (ER) is seen juxtaposed to a mitochondrial constriction with numerous microtubules crossing the junction at acute angles and wrapping around the mitochondria with a gentle twist (*Figure 1C*, *Figure 1—figure supplement 1*). In addition, large (~500 nm diameter) vesicles are seen in their entirety budding from near the junction, and lipid droplets are present in the vicinity (*Figure 1B,C*).

## Mitochondria contain dense, granular deposits in the matrix

In normative intracellular contexts such as those described above, we consistently observed dense granules within mitochondria in a variety of mammalian cell types (*Figure 1*, *Figure 1—figure supplements 1–2*, and *Videos 1–9*). *Figure 1D* shows granules dispersed between flat lamellar cristae and adjacent to well-formed junctions between cristae and the inner mitochondrial membrane. The presence of granules did not require mitochondrial proximity to the ER, nor did granule appearance correlate with mitochondrial shape (round or elongated) (*Figure 1E*). Granule diameters ranged from about 20 to 100 nm, varying with cell type. As expected, no granules were seen in fully constricted fission tubules, which had a minimum diameter of approximately 15 nm and from which matrix was excluded. Granules were seen in mitochondria both near the cell periphery (*Figure 1B*) and adjacent to the nucleus (*Figure 1F*). We note that dense spots of similar size have appeared without description or discussion in published cryo-TEM images (*Faas et al., 2012*; *Woodward et al., 2015*), supporting the notion that granules are widely found in mammalian mitochondria. Granules are particularly striking in CSTET data due to the greater depth range and higher sensitivity to elemental composition of this technique.

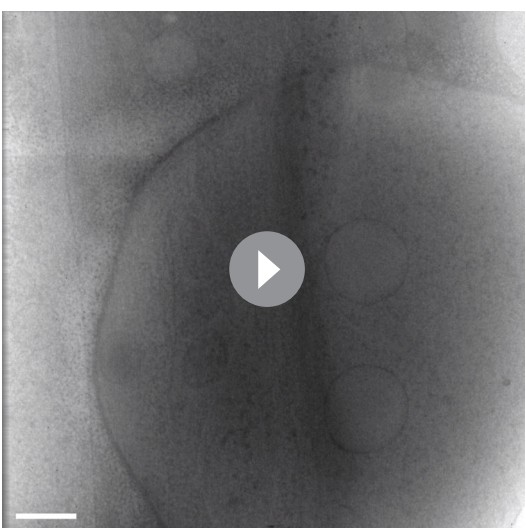

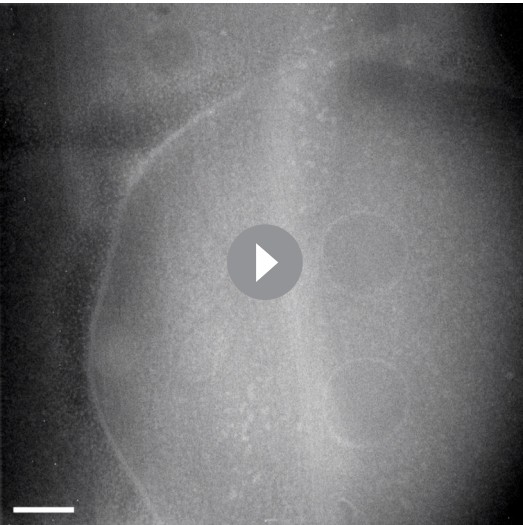

**Video 1.** Aligned tilt series of CSTET BF images from a region 750-nm thick within a WI-38 fibroblast. The tilt series corresponds to *Figure 1B*. Scale bar is 400 nm.
DOI: https://doi.org/10.7554/eLife.29929.005

**Video 2.** Aligned tilt series of CSTET DF images from a region 750-nm thick within a WI-38 fibroblast. The data for this tilt series were collected simultaneously with those shown in *Video 1*. Scale bar is 400 nm.
DOI: https://doi.org/10.7554/eLife.29929.006

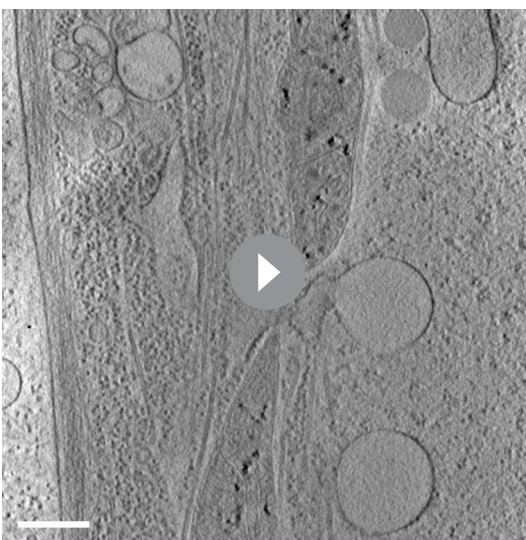

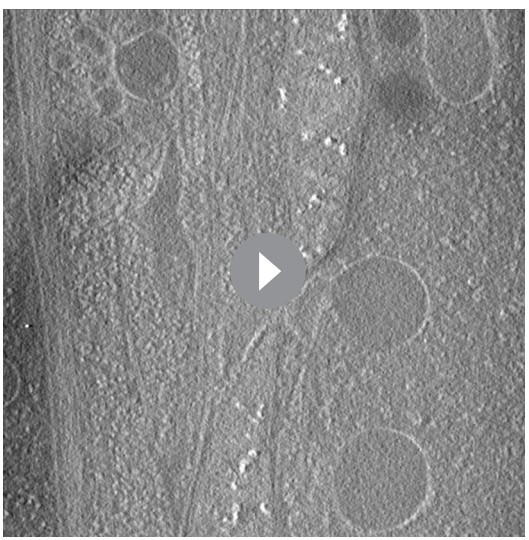

**Video 3.** BF tomographic reconstruction of a region within a WI-38 fibroblast. The reconstruction corresponds to *Figure 1B* and was done based on the tilt series shown in *Video 1*. Scale bar is 400 nm.
DOI: https://doi.org/10.7554/eLife.29929.007

**Video 4.** DF tomographic reconstruction of a region within a WI-38 fibroblast. The reconstruction was done based on the tilt series shown in *Video 2*.
DOI: https://doi.org/10.7554/eLife.29929.008

## Analysis of mitochondrial granule content and form

Previous descriptions and analyses of mitochondrial matrix granules were made using isolated and chemically fixed mitochondria (*Greenawalt et al., 1964*) or from observations of dehydrated or fixed and heavy metal-stained cellular material (*Bordat et al., 2004*; *Boonrungsiman et al., 2012*). Such manipulations introduce uncertainty regarding the reliability of the observations, and the nature of these granules and their physiological relevance remained unresolved. We therefore used the same fully hydrated, vitrified cells prepared for CSTET to analyze the elemental composition of

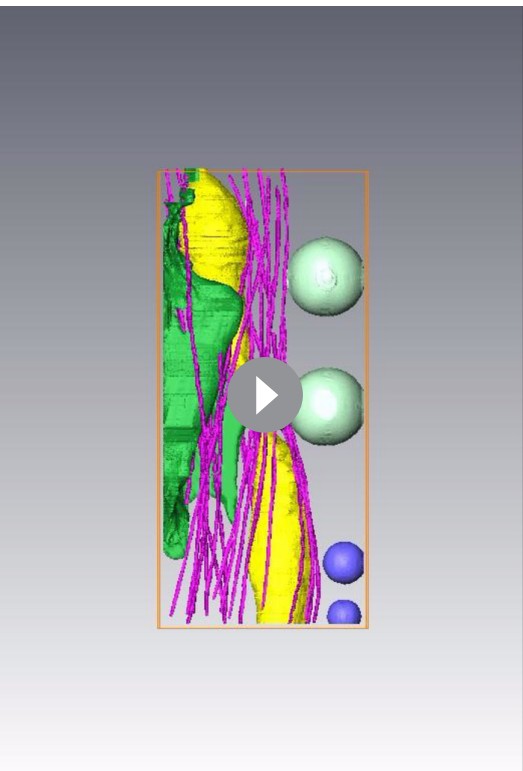

**Video 5.** Animation of segmentation shown in *Figure 1C*, overlaid on the reconstruction shown in *Figure 1B* and *Video 3*. The reconstruction contrast has been inverted.
DOI: https://doi.org/10.7554/eLife.29929.009

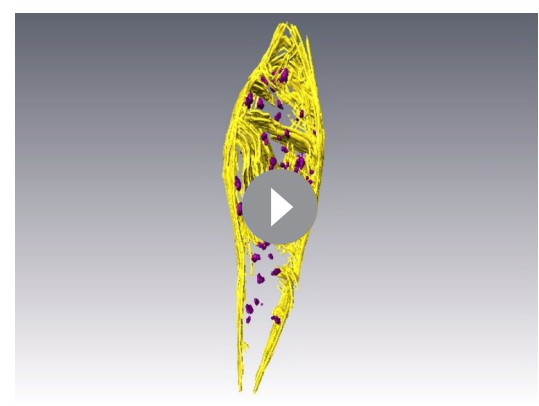

**Video 6.** Animation of segmentation shown in *Figure 1D*.
DOI: https://doi.org/10.7554/eLife.29929.010

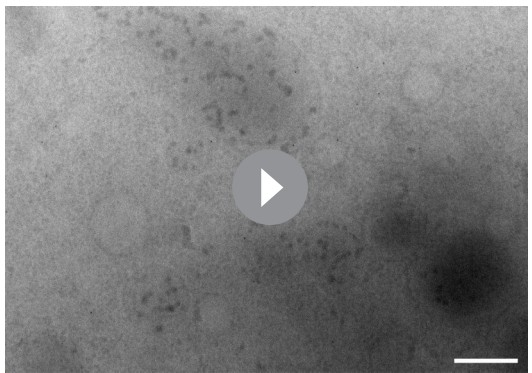

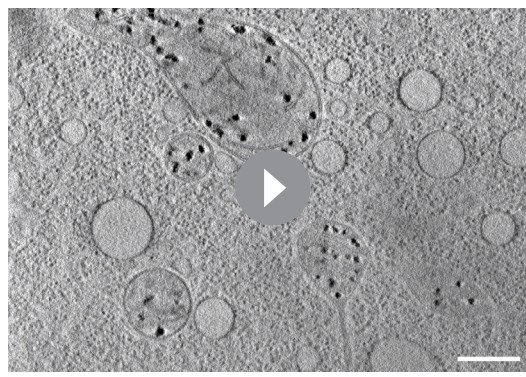

**Video 7.** Aligned tilt series of CSTET BF images from a region within a human dermal microvascular endothelial cell, shown in *Figure 1—figure supplement 2B*. Scale bar is 400 nm.
DOI: https://doi.org/10.7554/eLife.29929.011

**Video 8.** BF tomographic reconstruction of a region within a human dermal microvascular endothelial cell, shown in *Figure 1—figure supplement 2B*. The reconstruction was done based on the tilt series shown in *Video 7*. Scale bar is 400 nm.
DOI: https://doi.org/10.7554/eLife.29929.012

the granules. Energy-dispersive X-ray (EDX) spectroscopy gave signals for both calcium and phosphorus from granule-rich mitochondria but not from adjacent cytosolic regions (*Figure 2A*). Diffraction effects (blinking in the bright field) were not observed during the tilt series acquisitions for tomography, indicating that the granules were not crystalline. We conclude that mitochondrial matrix granules in intact cells are composed of amorphous calcium phosphate, as was surmised (*Lehninger, 1970*; *Chalmers and Nicholls, 2003*) but never directly demonstrated in samples not subjected to fixation or dehydration.

Complementing the spectroscopic characterization, the electron imaging experiment itself provided additional insight into granule form, composition, and density. STEM mode enables one to take advantage of the relation between atomic number and the angular distribution of electron scattering; bright-field (BF) images are generated from electrons transmitted in the forward direction, while dark-field (DF) images are formed from electrons scattered to higher angles in the same experiment (*Wolf et al., 2014*) (*Figure 2B*). Strong high-angle scattering by mitochondrial granules (i.e. black appearance in the BF image and white in the DF) (*Figure 2C*) is consistent with a composition biased toward elements heavier than the oxygen dominating the surrounding aqueous medium.

Ribosomes (*Figure 2D*) also appear dark in the BF and are moderately bright in the DF due to their density and content of relatively heavy elements, including phosphorus. For comparison, lipid droplets (*Figure 2E*) are characteristically dark in both the BF and the DF images due to their dense composition of the light element carbon. In addition to containing granules, mitochondrial matrices are darker in the BF than the surrounding cytosol (*Figures 1* and *2*), suggesting the presence of dissolved heavy ions. Due to the high background in fully hydrated cells, we were not able to obtain sufficient EDX signal from mitochondrial regions lacking granules to compare soluble mitochondrial ion content to cytosol using this method. This limitation was not surprising, considering that the concentration of matrix calcium ions would reach a maximum of only 100 mM at electrochemical equilibrium, and actual steady-state levels are considerably lower (*Pozzan et al., 2000*), whereas the

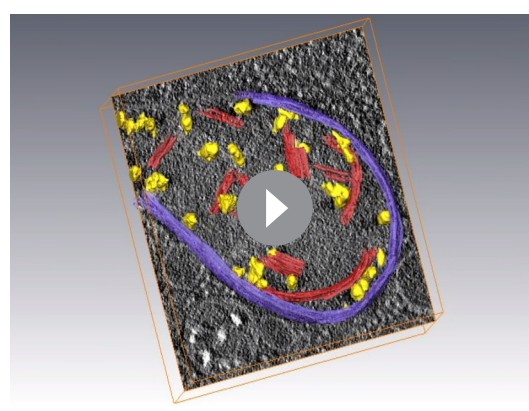

**Video 9.** Animation of segmentation shown in *Figure 1—figure supplement 2C*, overlaid on the reconstruction shown in *Figure 1—figure supplement 2B* and *Video 8*. The BF reconstruction contrast has been inverted.
DOI: https://doi.org/10.7554/eLife.29929.013

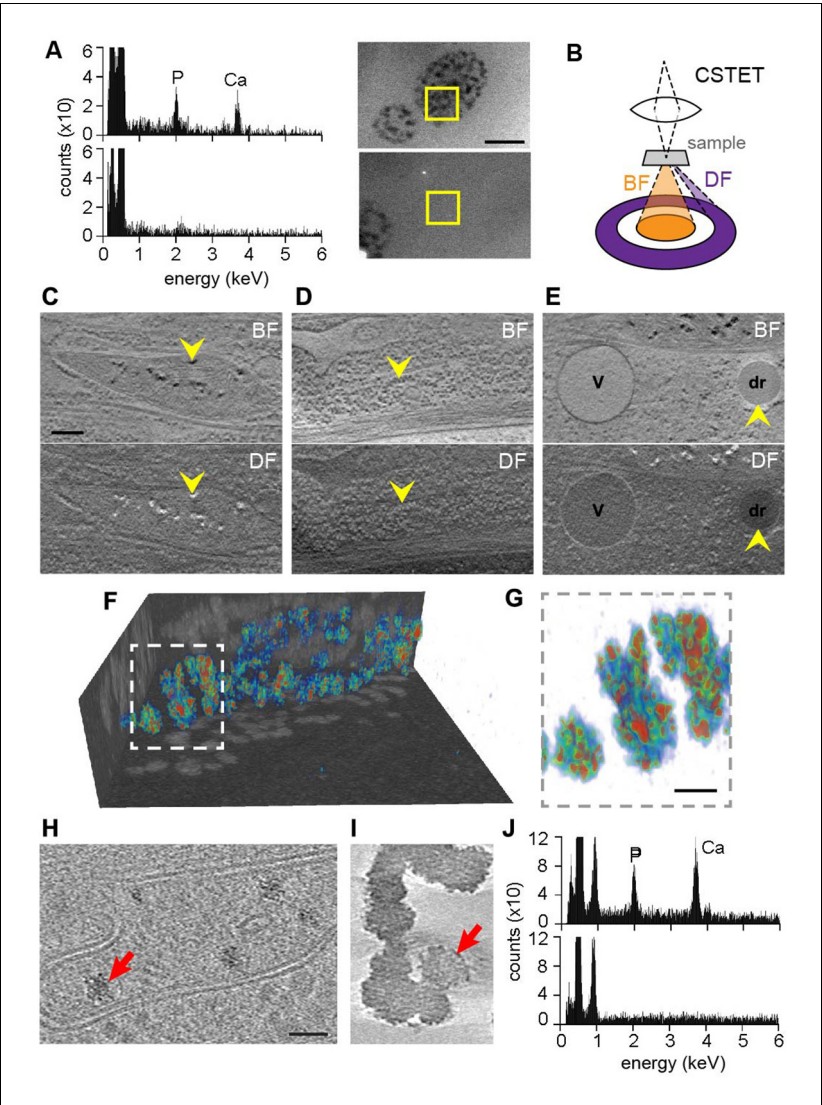

**Figure 2.** Elemental characterization of mitochondrial granules. (**A**) EDX identifies calcium (Ca) and phosphorus (P) enrichment in mitochondria. Areas of a WI-38 fibroblast subjected to EDX (boxed) were imaged prior to spectroscopic analysis. Scale bar is 400 nm. (**B**) Schematic (not to scale) showing collection of BF and DF CSTET data. (**C–E**) Scale bar is 200 nm. Sections of BF (top) and DF (bottom) reconstructions display (**C**) mitochondrial granules (arrowheads), (**D**) a polyribosome (arrowheads), and (**E**) a lipid droplet (dr; arrowheads) and a vesicle (v). (**F**) Color-coded 3D volume rendering (red: high density; blue: low density) showing heterogeneity of density in granules of a WI-38 fibroblast. Granules are presented against backdrops of projected BF volume densities with inverted contrast. (**G**) Zoom in on the boxed region of F. Scale bar is 50 nm. Sections 10-nm thick from zlTEM tomographic reconstructions are shown for (**H**) a thin region of a WI-38 fibroblast cell displaying mitochondrial granules and (**I**) synthetic amorphous calcium phosphate. The synthetic particles shown here were obtained 1.5 min after mixing of calcium and phosphate solutions. Red arrows in panels H and I highlight particles for comparison. Scale bar in H is 50 nm and applies also to I. (**J**) EDX of synthetic amorphous calcium phosphate (top) compared to an adjacent region of vitrified solution (bottom).

DOI: https://doi.org/10.7554/eLife.29929.014

The following figure supplements are available for figure 2:

**Figure supplement 1.** Intensity thresholding for quantitative estimation of mitochondrial granule scattering.
DOI: https://doi.org/10.7554/eLife.29929.015

**Figure supplement 2.** Large granules are observed in mitochondria of murine embryonic fibroblasts (MEFs) and are independent of the presence of the mitochondrial calcium uniporter MCU.
DOI: https://doi.org/10.7554/eLife.29929.016

concentrations of ions in solid deposits are much higher, as shown below. The intensity differences seen in CSTET reconstructions of mitochondrial matrix compared to cytosol therefore provide information difficult to obtain using other techniques.

The substantial heterogeneity of intensities observed in 3D within individual matrix granules (*Figure 2F*) is consistent with a composition of 'micro-packets' of solid calcium phosphate (*Lehninger, 1970*). It remains to be determined whether proteins or metabolites associate with these packets to regulate their growth and dissolution. For comparison with granules formed in mitochondria, biomimetic amorphous calcium phosphate particles were prepared in solution as described (*Habraken et al., 2013*). These particles, embedded in a thin film of vitreous ice, were compared with mitochondrial granules in a particularly thin region of a fibroblast cell (220 nm) using zero-loss energy filtered cryo-TEM (zlTEM) tomography recorded on a direct electron detector in movie mode. Mitochondrial and synthetic particles both display granularity and appear to be composed of smaller subparticles of about 4 nm in diameter (*Figure 2H,I*). The general features and appearance of the mitochondrial and synthetic granules are similar, as are the EDX spectra (*Figure 2A,J*).

Voxel intensities in STEM reconstructions were used to obtain quantitative information on the mass densities within matrix granules, using the known composition of ribosomes as a reference. The most strongly scattering regions of the matrix granules in CSTET experiments were estimated to be 34–48% the density of crystalline tricalcium phosphate (TCP) (*Figure 2—figure supplement 1*; *Tables 1–3*). Granules occupied up to 20% of mitochondrial volumes of mouse embryonic fibroblasts (MEFs) (*Figure 2—figure supplement 2*). If solubilized within the matrix volume, granule material would correspond to molar calcium and phosphate concentrations, showing that mitochondria in living cells sequester substantial ion reserves in solid form.

## The effect of cell stress on mitochondrial matrix granules

We next explored how perturbations to cell homeostasis and mitochondrial function affect calcium sequestration. Due to previous association of mitochondrial granules with pathology (*Webster, 2000*; *Dong et al., 2006*), we tested the effect of induced cell stress on mitochondrial matrix granules. After treatment of human primary embryonic lung fibroblasts with the chemotherapeutic agent doxorubicin, CSTET revealed dramatically increased organelle fragmentation and formation of autophagosomes/autolysosomes (*Figure 3A*), as well as instances of mitochondrial aggregation (*Figure 3B*), reported to occur in early-stage apoptosis (*Haga et al., 2003*). Matrix granules in doxorubicin-treated cells (*Figure 3A,B*) were generally similar to those in mitochondria of untreated cells (*Figure 1B,D,E*). Interestingly, we visualized in a doxorubicin-treated cell two mitochondria only 700 nm apart but in distinct microenvironments: one associated with cytoskeletal elements in the cytosol and the other sequestered with highly fragmented ER remnants, as if in the process of being decommissioned. Notably, these two mitochondria showed striking differences in granule size and density (*Figure 3C*), likely reflecting their distinct energetic states and diverging fates. Both the large and small granules consisted of amorphous material. In contrast, in a different cell region showing particularly severe damage and organelle degradation, structures that may have been fragmented mitochondria contained dense clusters of fibers or crystalline needles (*Figure 3—figure supplement 1*). The forces or factors that preserve the normal organization of material within matrix granules may have dissipated in this case.

## Granule formation requires mitochondrial polarization but not the calcium uniporter

Embryonic fibroblasts from knockout mice lacking the mitochondrial calcium uniporter (MCU) (*Pan et al., 2013*) showed deposits indistinguishable from control cells and containing calcium and phosphorus as demonstrated by EDX analysis (*Figure 2—figure supplement 2*), supporting the existence of other mechanisms for calcium entry (*Elustondo et al., 2017*). Calcium uptake depends on the mitochondrial membrane potential (*Szabadkai and Duchen, 2008*), and the dye JC-1, commonly used as a membrane potential reporter, enabled visualization of mitochondria by cryo-fluorescence followed by CSTET (*Figure 4*). However, JC-1 damaged mitochondrial ultrastructure and disrupted granules, likely by perturbing the potential it reports on. This experiment demonstrated the feasibility of correlative fluorescence/CSTET studies but also emphasizes that fluorescent sensors may perturb organelle chemistry and morphology, effects that are readily seen in CSTET. Intentional

depolarization of mitochondria by treating cells with carbonyl cyanide p-[trifluoromethoxy]-phenyl-hydrazone (FCCP) eliminated calcium deposits (*Figure 5A,B,C*), demonstrating the relevance of the proton gradient to matrix granule formation in cells. As granules were found consistently in untreated non-confluent fibroblasts and thus presumably existed before FCCP treatment, this experiment also showed that ion deposition is reversible, though the rate of mobilization remains to be determined. The matrices of some FCCP-treated mitochondria showed contrast similar to cytosol, indicating depletion of soluble heavy elements as well (*Figure 5A,B,C*). Remarkably, neighboring mitochondria, including some sharing contiguous outer membranes (*Figure 5C*), displayed very different matrix densities, revealing that variations in soluble ion concentrations, and not only the presence of dense deposits, can be detected by CSTET.

Overall, granules were observed in hundreds of mitochondria imaged in this study in multiple adherent mammalian cell types visualized under sub-confluent conditions. However, certain observations suggested situations that may deplete solid calcium stores. In one instance, we observed granule-free mitochondria in MCF10A breast epithelial cells that were undergoing a burst of proliferation at the time of vitrification (*Figure 5D*). It was difficult to visualize these cells when rapidly proliferating because the dense cell clusters that formed rarely presented regions 1 µm or less in thickness. Nevertheless, an effort was made to image an MCF10A cell on the edge of a cluster. A group of mitochondria in this case were observed to display the dark matrix observed in BF images of other mitochondria, suggesting that they remain polarized and have a high content of soluble calcium, but they were clearly devoid of granules. Mitochondria from other cells in the same culture were rich in granules (*Figure 5E*). It therefore appears that sequestration and mobilization of calcium occurs according to local signaling and energetic requirements and can exhibit substantial heterogeneity within certain cell populations.

We also observed that WI-38 fibroblast cells grown for longer periods and allowed to reach 90% to 95% confluence on grids often contained mitochondria with few or no granules (*Figure 5—figure supplement 1*). Again, imaging fibroblasts near confluence is challenging, as large fractions of the cultures were too thick even for CSTET, but certain areas were still accessible. Notably, confluent fibroblasts are highly metabolically active, using substantial energy to produce and secrete large amounts of extracellular matrix precursors (*Lemons et al., 2010*). Indeed, substantial extracellular matrix was detected in the fibroblast cultures imaged under these conditions (not shown).

## Conclusions

Establishing the proton gradient and synthesizing ATP are textbook functions of mitochondria, for which the chemiosmotic hypothesis is the accepted mechanism. By contrast, the observation that calcium uptake by mitochondria may take precedence over ATP synthesis (*Bernardi, 1999*) is not widely appreciated. Formation of solid calcium phosphate complexes in mitochondria has been described in the context of bone formation (*Boonrungsiman et al., 2012*; *Sutfin et al., 1971*; *Kerschnitzki et al., 2016*), but our study points to broader purposes for mitochondrial granules, beyond calcification. CSTET visualization of calcium sequestration in a variety of unfixed and unstained whole cells strengthens the notion that solid-state storage must be considered in a general accounting of cellular calcium, even for non-mineralizing cells. This direct and readily appreciable view of calcium deposition in intact cells is valuable for re-focusing attention on the role of ion phase separation and remobilization in mitochondrial function. Suggestions of granule depletion in highly metabolically active cells should be further investigated. Furthermore, quantitative analysis of electron scattering using CSTET data should be further developed to provide information on elemental content and concentrations that cannot readily be obtained by other methods. With the remarkable ability to reveal organelle organization and elemental distribution in intact mammalian cells with no sample manipulation other than cryo-preservation, CSTET promises to be a pioneering technology for cell biology.

## Materials and methods

**Key resources table**

| Reagent type (species) or resource | Designation | Source or reference | Identifiers |
| --- | --- | --- | --- |

*Continued on next page*

*Continued*

| Reagent type (species) or resource | Designation | Source or reference | Identifiers |
|---|---|---|---|
| Cell line | WI-38 | Coriell Institute | NA06814 |
| Cell line | Human Dermal Microvascular Endothelial Cells (HDMEC) juvenile foreskin | Promocell | C-22020 |
| Cell line | U2OS | ATCC | HTB-96 |
| Cell line | MCF10A | ATCC | CRL-10317 |
| Cell line | mouse embryonic fibroblasts | Weizmann Institute Stem Cell Core Unit | MEFs |
| Cell line | mouse embryonic fibroblasts (MCU -/-) | *Pan et al., 2013* | MEFs MCU -/- |
| Cell line | mouse embryonic fibroblasts (MCU +/+) | *Pan et al., 2013* | MEFs MCU +/+ |

## Cell culture and sample preparation

WI-38 embryonic lung fibroblasts were purchased from Coriell and maintained in Minimal Essential Medium (Gibco) supplemented with 15% fetal calf serum, L-glutamine, and penicillin/streptomycin. MEFs were obtained from the Stem Cell Core Unit of the Weizmann Institute and maintained in Dulbecco's modified Eagle's medium (DMEM; Gibco) with 10% fetal calf serum, L-glutamine, and penicillin/streptomycin. MEFs from MCU -/- mice and MCU +/+ littermates were obtained from the laboratory of Toren Finkel at NIH and maintained in the same medium. Cell cultures grown in parallel to those placed on grids were genotyped by PCR using the primers shown in *Figure 2—figure supplement 2*. These primers produce for MCU +/+ mice an expected amplicon of 190 bp that is eliminated in MCU -/- mice by insertion of the trapping vector (*Pan et al., 2013*). MCF10A breast epithelial cells, obtained from the laboratory of Yosef Yarden at the Weizmann Institute, were grown in DMEM:F12 medium (Biological Industries) supplemented with 5% horse serum, L-glutamine, penicillin/streptomycin, 10 µg/ml insulin, 10 ng/ml EGF (Peprotech), 0.5 µg/ml hydrocortisone (Sigma), and 100 ng/ml cholera toxin (Sigma). U2OS cells, obtained from the laboratory of Benjamin Geiger at the Weizmann Institute, were maintained in DMEM (Gibco) supplemented with 10% fetal calf serum, 5% L-glutamine, and penicillin/streptomycin. Human dermal endothelial cells (HDMEC) supplied by Promocell (Heidelberg, Germany) were obtained from the laboratory of Ronen Alon at the Weizmann Institute and grown in the endothelial cell growth medium purchased from Promocell. WI-38 and MCF10A cell cultures were tested for mycoplasma every 3 weeks.

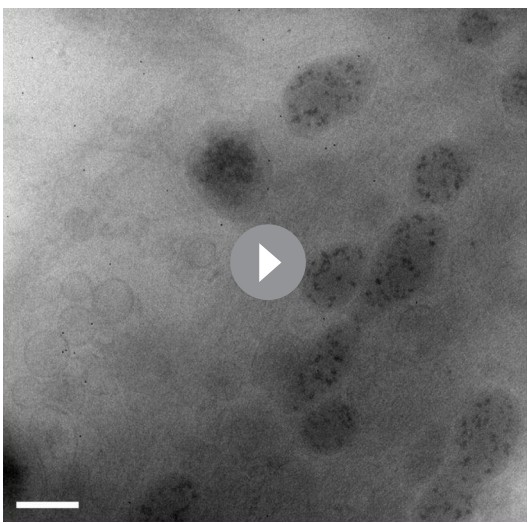

**Video 10.** Aligned tilt series of CSTET BF images from a region within a WI-38 fibroblast treated with doxorubicin. Scale bar is 400 nm.
DOI: https://doi.org/10.7554/eLife.29929.022

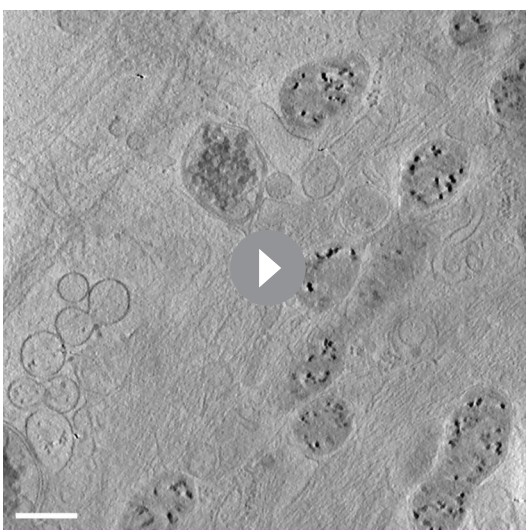

**Video 11.** BF tomographic reconstruction of the tilt series in *Video 10*. Scale bar is 400 nm.
DOI: https://doi.org/10.7554/eLife.29929.023

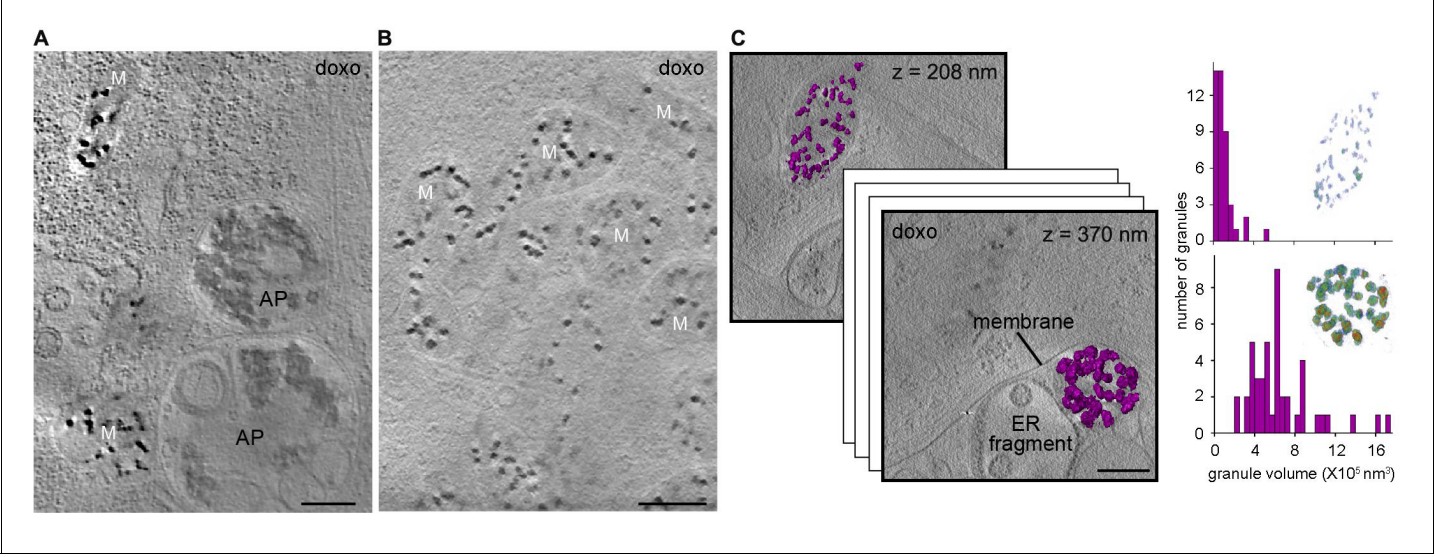

**Figure 3.** Effect of cell stress on mitochondrial granules. Sections 30-nm thick from BF CSTET reconstructions of doxorubicin-treated WI-38 cells are displayed. Scale bars are 400 nm. (**A**) Mitochondria in the vicinity of autophagosomes (AP) (cell is 890-nm thick in this region). (**B**) Aggregated mitochondria (cell is 1-µm thick). (**C**) Different granule sizes in two nearby mitochondria separated by a membrane. Granule volumes were segmented using an intensity threshold and are displayed in purple. Height (in nm) from the bottom of the cell (which is 800-nm thick in this region) is in the upper right corner of each section. Histograms of granule sizes are displayed for the two mitochondria. Insets show granules colored as in *Figure 1F*. Tilt series and reconstruction movies of a doxorubicin-treated cell are in *Videos 10* and *11*.

DOI: https://doi.org/10.7554/eLife.29929.020

The following figure supplement is available for figure 3:

**Figure supplement 1.** Region of a doxorubicin-treated fibroblast cell.

DOI: https://doi.org/10.7554/eLife.29929.021

Gold Quantifoil R3.5/1 grids were subjected to glow-discharge treatment and immediately immersed in water by placing them carbon-side up onto a glass (fibroblasts) or tissue-culture treated plastic (other cell types) coverslip affixed to the bottom of a tissue culture dish to facilitate manipulation of grids with surgical tweezers. Grids in the open tissue culture dishes filled with water were

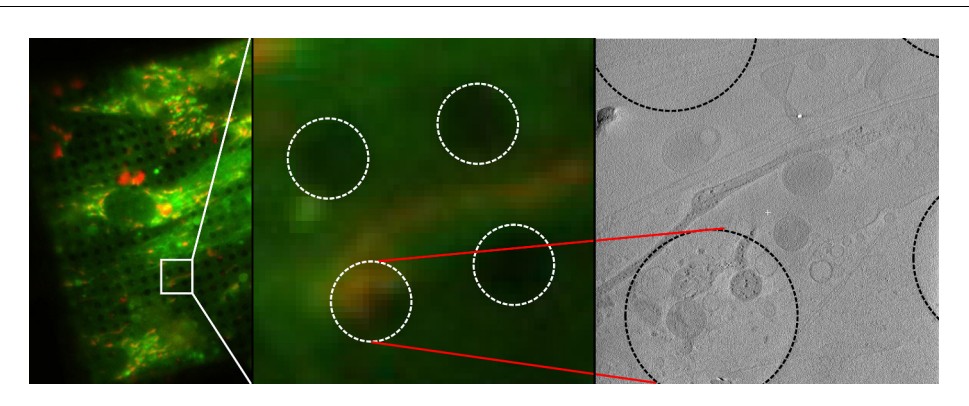

**Figure 4.** Correlative imaging of mitochondria using fluorescence from a membrane-potential reporter. Mitochondria in WI-38 fibroblasts were stained with JC-1, cryo-preserved, and imaged successively by fluorescence microscopy and CSTET. A field of JC-1 stained cells is shown on the left. The section of the field corresponding to the tomogram is enlarged in the middle panel. The right panel shows a slice of the tomogram. Dashed circles indicate the holes (3.5 µm diameter) in the carbon support on which the cells were grown. Mitochondria show irregular morphology and partial dissolution of granules.

DOI: https://doi.org/10.7554/eLife.29929.024

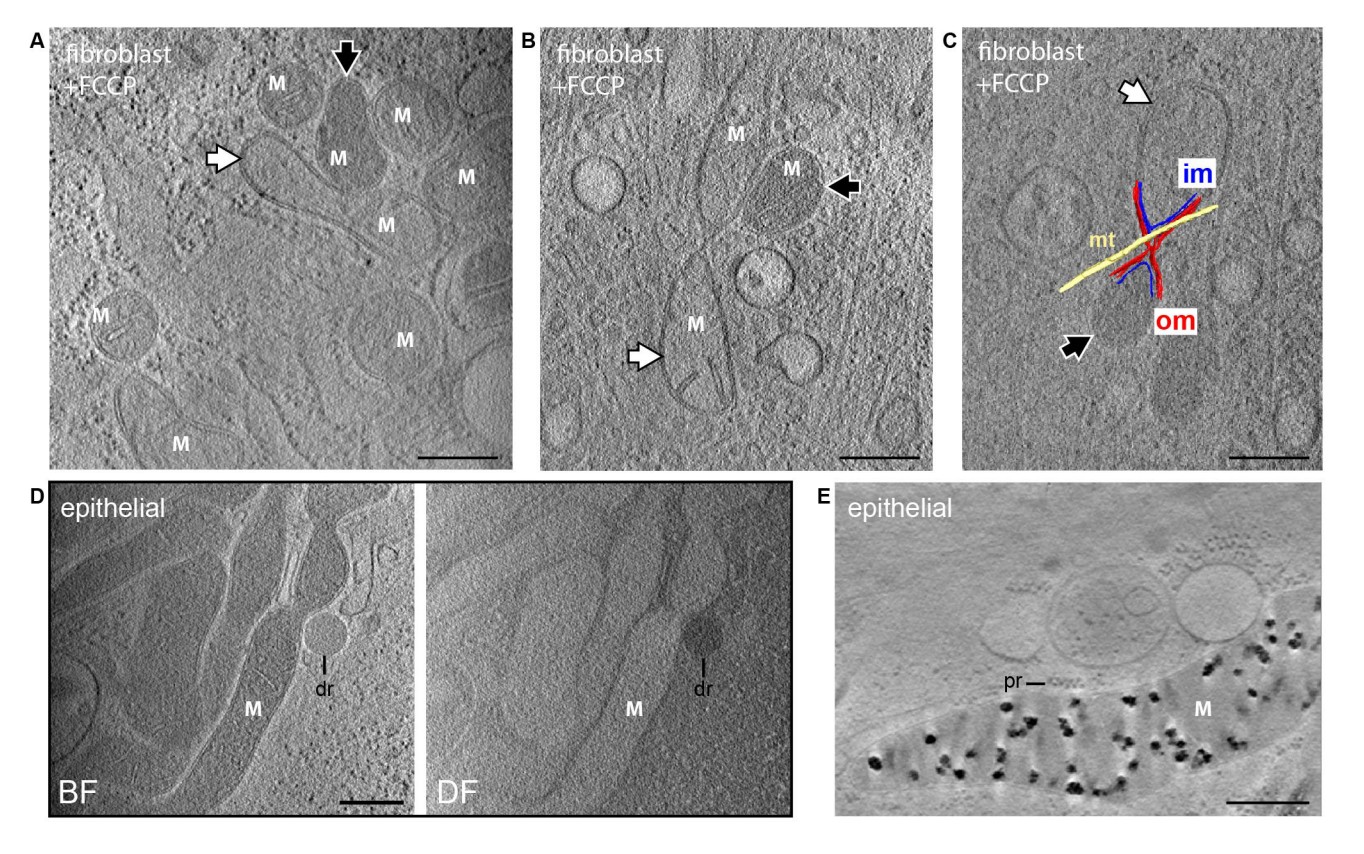

**Figure 5.** Dissipation of matrix granules. All panels show 30-nm thick sections from CSTET reconstructions. White M indicates mitochondria. Scale bars are 400 nm. (**A,B**) Fibroblasts treated with FCCP show no granules. White and black arrows indicate mitochondria with weakly and strongly scattering matrices, respectively. (**C**) Distinct matrix densities in mitochondria sharing contiguous outer membranes in FCCP-treated cells. om, outer membrane; im, inner membrane; mt, microtubule. (**D**) Group of granule-free mitochondria in MCF10A cells. dr, lipid droplet. Cell is 800-nm thick in this region. (**E**) MCF10A mitochondria containing granules. pr, polyribosome.

DOI: https://doi.org/10.7554/eLife.29929.025

The following figure supplement is available for figure 5:

**Figure supplement 1.** Granule-free mitochondria in fibroblasts nearing confluence.

DOI: https://doi.org/10.7554/eLife.29929.026

then UV sterilized for half an hour in a tissue culture hood. Grids for MCF10A, U2OS, and HDMEC cells were coated with fibronectin after sterilization. For all cell types, water was replaced with the appropriate culture medium, and cells were plated onto the grids. Cultures on grids were grown to 30–70% confluence, typically over 2 or 3 days. Immediately prior to vitrification, grids were lifted from the coverslip platform in the tissue culture dish, 1 µl colloidal gold (10 nm) (*Duchesne et al., 2008*) at a concentration of 120 nM in phosphate buffered saline was added to provide fiducial markers, and 4 µl of cell culture medium at 37°C was placed on the cell side of the grid. The grids were blotted at 21°C and >90% humidity for 2–4 s from the side opposite the cells, flash-frozen in liquid ethane using a Leica EM-GP plunger, and stored in liquid nitrogen until use.

FCCP (Sigma) was prepared as a 1 mM stock in DMSO and diluted 1:1000 into the medium of WI-38 fibroblasts cultured on grids to give a final concentration of 1 µM. Cells were grown for a further 24 hr before vitrification. Doxorubicin (Sigma) was prepared as a 2 mM stock in DMSO and diluted 1:2000 into WI-38 fibroblasts cultured on grids to give a final concentration of 1 µM. Samples were vitrified 16 hr after doxorubicin treatment. JC-1 was prepared as a 1 mg/ml stock in DMSO and diluted 100-fold into warm medium, which was then mixed 1:1 with the medium of cells cultured on grids. After 30-min incubation, cells were washed with PBS and vitrified.

The following is an accounting of the number of tomograms collected and the number of mitochondria observed for the different cell types and treatments. Mitochondria were counted as distinct if they did not share matrix content. During the course of this study, about 22 tomograms of untreated WI-38 fibroblasts that had grown 2 to 4 days on grids were collected. Altogether 66 mitochondria were observed in reconstructions of these tomograms, and all these mitochondria contained granules. In addition, 13 tomograms were collected from WI-38 fibroblasts that had grown for 6 or 7 days on grids without exchange of medium and were near confluence. Thirty mitochondria were observed in these tomograms, 15 of which contained granules. Seven tomograms of wild-type MEFs yielded 33 mitochondria, all containing granules, whereas 3 tomograms of MCU-/- MEFs showed 17 mitochondria, 7 of them lacking granules). Two tomograms of U2OS (eight mitochondria, all with granules), 6 tomograms of HDMEC (26 mitochondria, all with granules) and 3 tomograms of MCF10A cells (14 mitochondria, 7 of which had no granules) were acquired. Nine tomograms of WI-38 (59 mitochondria, 10 of which had no deposits) were taken after doxorubicin treatment, and 8 tomograms (68 mitochondria, none of which contained deposits) were taken after FCCP treatment. In addition to full tomograms, numerous single STEM images of cell regions were taken during this study. Due to the time investment in collecting tomograms, cell regions were not chosen randomly for tomography but were rather selected for ideal depth (600 nm to 1 μm) and for the likelihood of containing mitochondria, as assessed by the presence in single STEM images of organelles that appeared slightly darker than bulk cytosol.

## Cryo-light microscopy

Fluorescence imaging of JC-1 treated cells cryopreserved on Quantifoil grids was done using an Olympus BX51 microscope equipped with a cryostage (Linkam). LED excitation was at 470 nm. Emission was collected between 515 and 555 nm for the green channel and between 575 and 635 for the red channel. After fluorescence imaging, grids were stored in liquid nitrogen for subsequent electron microscopy.

## Cryo-scanning transmission electron tomography

Vitrified cell samples were observed with a Tecnai F20 S/TEM instrument at 200 kV. STEM was performed with extraction voltage = 4300 V, gun lens = 3 or 6, and spot size = 5, with 10 or 20 μm condenser apertures, yielding probe diameters of 2 or 1 nm and semi-convergence angles of 1.3 or 2.7 mrad, respectively. The camera length was set either to 320 or to 520 mm so that the acceptance cone semi-angle at the bottom-mounted bright-field detector (BF; Gatan model 805, later upgraded to 806) was slightly larger than the illumination cone semi-angle, providing conditions for on-axis BF signal (*Rez et al., 2016*). Simultaneous dark-field data were collected on a Fischione HAADF detector located at the 35 mm port of the microscope column. The geometry imposes a gap between the outer cutoff angle of the BF and the inner cutoff angle of the HAADF detectors; the ratio between these angles is approximately 4. Images of $1024 \times 1024$ or $2048 \times 2048$ pixels were recorded with probe dwell times between 5 and 20 μs, yielding frame exposure times of approximately 10 to 20 s. Spatial sampling was set between 1 and 4 nm/pixel. Doses, measured as described previously (*Wolf et al., 2014*), were limited to 1–3 electrons/$Å^2$ per dwell spot. Single-axis tilt series were recorded using either FEI Xplore3D software or SerialEM (*Mastronarde, 2005*), with angular sampling from ±60° in 2° steps. Data collection schemes were either sweeping from −60° to +60°, or collecting from +20° to −60°, and then from +22° to +60°. Dynamic focus adjustment was employed to maintain focus conditions perpendicular to the tilt axis at all tilt angles.

## Tomogram reconstruction and segmentation

The tomographic tilt series were aligned using fiducial markers and reconstructed using weighted back projection (*Frangakis and Hegerl, 2001*) as implemented in the IMOD software suite (*Kremer et al., 1996*). Reconstructions are displayed after non-linear anisotropic diffusion filtering within IMOD. Tomograms will be deposited in the Electron Microscopy Data Bank. Segmentation and volume rendering were performed using Amira 6.3 (FEI Visualization Sciences Group).

## EDX measurements

Energy-dispersive X-ray spectroscopy was performed in STEM mode on vitrified cell samples with the same electron microscope set-up as used for CSTET, using a liquid $N_2$ cooled Si(Li) detector (EDAX).

## Preparation and analysis of amorphous calcium phosphate

Amorphous calcium phosphate was prepared according to published protocol (*Habraken et al., 2013*). Briefly, 10 mM stocks of calcium chloride and potassium phosphate were prepared in Tris-buffered saline and mixed. At 1.5 min after mixing and every 5 min following, 5 μl were removed, placed on Quantifoil Multi A copper 200 mesh grids, and vitrified. Samples at several timepoints were inspected using CSTET and zero-loss energy-filtered cryo TEM tomography.

## Zero-loss energy filtered cryo TEM tomography

Vitrified cell samples were observed with the same Tecnai F20 S/TEM instrument as used for CSTET, which is equipped with a Gatan Quantum 967 'special' energy filter and K2 summit direct electron detector. Experiments were performed at 200 kV, with extraction voltage = 4300 V, gun lens = 4, and spot size = 5, a condenser aperture of 30 μm, and a zero-loss energy slit of 20 eV. Dose was set to 0.12 electrons/Å$^2$/sec, with 9 s exposures divided over 18 frames in movie mode. Motion correction was performed using MotionCor2 (*Zheng et al., 2017*). The data collection scheme was the same as used for CSTET. TEM tilt series were corrected for CTF using phase-flipping routine in IMOD.

## TEM imaging of thin cell sections

For preparation of thin cell sections, WI-38 cells were grown on glass coverslips, fixed using 2.5% gluteraldehyde, 2% paraformaldehyde in 0.1 M sodium cacodylate buffer at room temperature, washed in cacodylate buffer at 4°C, and stained with 1% osmium tetroxide and 2% uranyl acetate. Samples were dehydrated in cold ethanol and then embedded in Epon. After removing the glass coverslips with liquid nitrogen, ultrathin sections of the monolayer Epon blocks (~70 nm) were cut parallel to the cover slip surface and transferred to 200-mesh copper grids. Thin plastic sections were imaged in a FEI Tecnai Spirit T-12 operated at a 120 kV. Images were recorded with a 2 k Eagle CCD camera (FEI, Eindhoven).

## Estimate of calcium/phosphate densities

For quantitative analysis, reconstructions were performed using the simultaneous iterative reconstruction technique (SIRT) as implemented in the Tomo3D program (*Agulleiro and Fernandez, 2015*). SIRT compares projections of the reconstructed volume with the raw data to minimize numerical differences. BF data, which showed greater signal-to-noise than DF data, were used for the analysis presented here. A band-pass filter with lower and upper cutoff of 1 and 100 pixel widths, respectively, was imposed to reduce minor shading artifacts.

The known composition of ribosomes and water were used to scale the image intensities. To estimate the expected scattering away from the BF detector by ribosomes, differential scattering cross-sections were integrated from the BF outer cutoff angle (five mrad) up to $\pi$ rad according to the densities of constituent elements (*Table 1*). In addition to the scattering of whole ribosomes, scattering of ribosomal RNA was calculated separately, since the RNA component will scatter more strongly than the protein and solvent components. The integrated cross-section for TCP with chemical formula $Ca_3(PO_4)_2$ and density 3.14 gm/cm$^3$ was calculated similarly.

Five intensity levels were measured from a representative tomogram: two for the granules (peak and inclusive levels), two for ribosomes (peak and inclusive levels), and one for cytosolic fluid, used to approximate water (*Table 2*). These intensities were determined by adjusting a threshold using ImageJ (*Schneider et al., 2012*) until the objects of interest were selected. Measured intensities in the tomographic reconstruction include an additive background $x$ from unscattered illumination, which can be estimated by imposing predicted intensity ratios between known components. For example, the ratio of ribosomal RNA (peak ribosomal intensities) (*Voss and Gerstein, 2005*) to water would be (5.3-$x$)/(16-$x$)=2.29, yielding $x$ = 24.3. The choice of total ribosomes (inclusive ribosome intensities) and the corresponding predicted intensity ratios, (7.5-$x$)/(16-$x$)=1.56, yields a value

**Table 1.** Computation of atom number densities for ribosomes and calcium phosphate.

| | Element | # of atoms | Mole fraction | # atoms/nm$^3$ |
|---|---|---|---|---|
| ribosomal RNA* | C | 68671 | 0.295 | 31.2 |
| | H | 78158 | 0.336 | 35.5 |
| | N | 27884 | 0.120 | 12.7 |
| | O | 50462 | 0.217 | 22.9 |
| | P | 7216 | 0.031 | 3.3 |
| | Mg | 239 | 0.001 | 0.24 |
| | | | | |
| Ribosomes[†] | C | 135061 | 0.186 | 19.3 |
| | H[‡] | 372416 | 0.514 | 53.2 |
| | N | 48041 | 0.0663 | 6.86 |
| | O[‡] | 161037 | 0.222 | 23.0 |
| | S | 501 | 0.000692 | 0.072 |
| | P | 7216 | 0.00996 | 1.03 |
| | Mg | 239 | 0.000330 | 0.034 |
| | | | | |
| $Ca_3(PO_4)_2$ | Ca | 3 | 0.231 | 18.4 |
| as crystalline TCP[§] | P | 2 | 0.154 | 12.2 |
| | O | 8 | 0.615 | 49.1 |

* The partial specific volume of RNA was taken to be 0.569 cm$^3$/g (**Voss and Gerstein, 2005**).

[†] A volume of 7000 nm$^3$ was estimated to enclose the ribosome (Protein Data Bank ID 4UG0) based on a ~ 2 nm-resolution isosurface calculated using Chimera (**Pettersen et al., 2004**). Solvent within this isosurface (42%) was treated as bulk vitreous ice for atom number density summations.

[‡] Includes solvent atoms.

[§] A density of 3.14 g/cm$^3$ was taken for crystalline TCP.

DOI: https://doi.org/10.7554/eLife.29929.017

of $x$ = 30.9. Using these background values and the predicted scattering for TCP, which is 6.06 times that of water, the expected intensity for TCP in tomogram images was calculated. On the scale from background to TCP thus obtained, the maximal and typical densities of the granules were found to be in the range of 34% to 48% and 31% to 42% of TCP density, respectively. The corresponding mass densities are 1.1 to 1.5 and 1.0 to 1.3 gm/cm$^3$, respectively. These values can be compared with a density of 2 gm/cm$^3$ for amorphous calcium phosphate (ACP) prepared in vitro (**Fawcett, 1973**).

For mitochondrial volume fraction estimates, MEF mitochondria were segmented in Chimera (**Pettersen et al., 2004**) applying conservative intensity thresholds to define granule boundaries. A rectangular prism was selected within a mitochondrion. The ratio of the sum of the volumes of the segmented granule regions within the rectangular prism to the volume of the prism was about 0.2, indicating that about 20% of the mitochondrial volume is occupied by granules in this case. Taking as a lower estimate a density 31% that of TCP (**Table 3**), a density of $3.2 \times 10^{-3}$ mol/cm$^3$ is obtained for calcium phosphate within the granules. The equivalent liquid concentration based on the 20% vol estimate would be about 0.64 M, corresponding to 1.9 M calcium ions and 1.3 M phosphate ions. These concentrations are compatible with the solubility of calcium ions (e.g. from $CaCl_2$) in aqueous solution but exceed by a factor of ~10$^4$ the solubility of $Ca_3(PO_4)_2$ in aqueous solution.

**Table 2.** Prediction of BF signals for conditions used in tomographic data collection.
From these data, we obtain the predicted intensity ratios between ribosomes and water, ribosomal RNA and water, and TCP and water.

| Scattering cross-sections per atom (>5 mrad) | |
| --- | --- |
| H* | - |
| C | 0.0042 |
| N | 0.0047 |
| O | 0.0051 |
| P | 0.0165 |
| Mg | 0.0097 |
| S | 0.0182 |
| Ca | 0.0278 |

| Estimated scattering signal per $nm^3$ material | |
| --- | --- |
| Water | 0.159 |
| Ribosome | 0.249 |
| ribosomal RNA | 0.364 |
| TCP | 0.963 |

| Predicted scattering intensity ratios | |
| --- | --- |
| Ribosome/water | 1.57 |
| RNA/water | 2.29 |
| TCP/water | 6.06 |

\* Hydrogen does not scatter electrons above the cutoff angle for the BF detector.
DOI: https://doi.org/10.7554/eLife.29929.018

**Table 3.** Granule density evaluation.

| Component | Threshold level in tomogram | Ratio to water | Ratio to TCP |
| --- | --- | --- | --- |
| Granule (peak) | 0.3 | 2.89 (2.05)[†] | 0.48 (0.34)[†] |
| Granule (inclusive) | 3.0 | 2.57 (1.87) | 0.42 (0.31) |
| Ribosome (peak) | 5.3 | 2.29 (1.72) | 0.38 (0.28) |
| Ribosome (inclusive) | 7.5 | 2.02 (1.57) | 0.33 (0.26) |
| Water (cytosol) | 16 | 1.00 | 0.16 |
| Extrapolated background | 24.3* (30.9)[†] | | |
| TCP[c] | −26.0[‡] (−59.4) | | |

\* Calculated using the threshold for peak ribosome densities and the predicted scattering ratio for ribosomal RNA vs. water given in **Table 2**.
[†] Values in parentheses were calculated using the threshold and predicted scattering intensity ratio for complete ribosomes vs. water given in **Table 2**.
[‡] The threshold for TCP is predicted based on the scattering intensity ratio given in **Table 2**.
DOI: https://doi.org/10.7554/eLife.29929.019

## Acknowledgements

We thank N Elad from the Department of Chemical Research Support for assistance with sample vitrification and T Finkel (NIH) for providing fibroblasts from MCU knockout mice. This study was supported by the European Research Council under the European Union's Seventh Framework Programme, grant number 310649 to D Fass, and by the I-CORE Program of the Planning and Budgeting Committee and The Israel Science Foundation (grant No. 1775/12). The work was also supported by Israel Science Foundation grant 1285/14 to M Elbaum and SG Wolf and by the Irving and Cherna Moskowitz Center for Nano and Bio-Nano Imaging at the Weizmann Institute of Science.

## Additional information

### Funding

| Funder | Grant reference number | Author |
| --- | --- | --- |
| Israel Science Foundation | 1285/14 | Sharon Grayer Wolf<br>Michael Elbaum |
| European Research Council | 310649 | Deborah Fass |
| Israeli Centers for Research Excellence | 1775/12 | Deborah Fass |

The funders had no role in study design, data collection and interpretation, or the decision to submit the work for publication.

### Author contributions

Sharon Grayer Wolf, Deborah Fass, Conceptualization, Data curation, Formal analysis, Supervision, Funding acquisition, Investigation, Visualization, Methodology, Writing—original draft, Project administration, Writing—review and editing; Yael Mutsafi, Conceptualization, sample preparation, Formal analysis, Investigation Methodology; Tali Dadosh, Formal analysis, Visualization, Methodology, Writing—review and editing; Tal Ilani, Ben Horowitz, Sarah Rubin, Sample preparation, Methodology, Writing—review and editing; Zipora Lansky, Sample preparation, Methodology; Michael Elbaum, Conceptualization, Formal analysis, Funding acquisition, Investigation, Visualization, Methodology, Writing—review and editing

### Author ORCIDs

Sharon Grayer Wolf  http://orcid.org/0000-0002-5337-5063
Michael Elbaum  https://orcid.org/0000-0001-7915-5512
Deborah Fass  http://orcid.org/0000-0001-9418-6069

### Decision letter and Author response

Decision letter https://doi.org/10.7554/eLife.29929.028
Author response https://doi.org/10.7554/eLife.29929.029

## Additional files

### Supplementary files

• Transparent reporting form
DOI: https://doi.org/10.7554/eLife.29929.027

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
