## [Decision Letter]

Thank you for submitting your article "3D visualization of mitochondrial solid-phase calcium stores in whole cells" for consideration by *eLife*. Your article has been reviewed by two peer reviewers, and the evaluation has been overseen by Werner Kühlbrandt as the Reviewing Editor and Andrea Musacchio as the Senior Editor. The reviewers have opted to remain anonymous.

The reviewers have discussed the reviews with one another and the Reviewing Editor has drafted this decision to help you prepare a revised submission.

Summary:

This is a very interesting manuscript using an advanced method to investigate the amount and distribution of Ca^2+^ in mitochondrial matrix (CSTET). The authors have advanced CSTET as a useful approach to imaging frozen, unstained cells of greater thickness than specimens accessible to more widely applied electron cryotomography methods. The present study is therefore an important application. The combination of BF and DF shows a sensitivity to atomic number. In their images, the darkness of the mitochondrial matrix itself is attributed to ion concentration. Furthermore, the authors have applied EDX spectroscopy to identify the granule content.

By the CSTET approach, the authors demonstrate that mitochondria contain molar Ca^2+^ concentrations and postulate that their participation in Ca^2+^ buffering depends on the phase separation between soluble and complexed Ca^2+^. The concept of amorphous Ca-Pi precipitates in the mitochondrial matrix dates back to the sixties. However, this is the first direct measurement of these stores, together with their quantification. The discovery of such important Ca^2+^ reservoirs, the fact that they appear to be MCU independent but dependent on membrane potential opens new avenues in cell biology and mitochondrial research.

Essential revisions:

1) The demonstration of MCU independence for granule formation is strong, but is there anything known about what happens to these granules in the absence of the Pi carrier, or when it is mutated?

2) Are these stores also retrieved in *S. cerevisiae* that lacks MCU and where the ER is not a Ca^2+^ store?

3) The demonstration of calcium and phosphorus signal by EDX for the mitochondrion shown in Figure 2 is convincing. But given that the authors also observe increased density for the mitochondrial matrix compared to cytoplasm by CSTET and attribute this to ions, the study should show the EDX signal for a mitochondrion that does not possess dense granules. Also, can imaging and spectroscopy be performed directly on amorphous calcium phosphate for comparison?

4) In reporting observations on granule presence and distribution in normal cells and after treatment with doxorubicin or FCCP, the authors should give the number of mitochondria imaged and the number with/without granules or specific granule variations (e.g. size variation in Figure 3). This is the only way a reader can assess whether the changes shown in the image examples are really of significance.

5) The importance of the JC-1 staining experiment to the paper overall should be made clear.

6) The number of replicates of experiments should be reported directly in the manuscript and not just in the *eLife* transparent report form.

---

## [Author Response]

[…] By the CSTET approach, the authors demonstrate that mitochondria contain molar Ca^2+^ concentrations and postulate that their participation in Ca^2+^ buffering depends on the phase separation between soluble and complexed Ca^2+^. The concept of amorphous Ca-Pi precipitates in the mitochondrial matrix dates back to the sixties. However, this is the first direct measurement of these stores, together with their quantification. The discovery of such important Ca^2+^ reservoirs, the fact that they appear to be MCU independent but dependent on membrane potential opens new avenues in cell biology and mitochondrial research.

Though the reviewers did not request it, we have added to the manuscript two pieces of information relating to the MCU knockout:

1) We now supply as supplementary information the results of PCR genotyping of the MCU knockout (MCU -/-) and control cells used for the analysis.

2) We have added to the supplementary information EDX analysis of MCU -/- mitochondria to confirm that the granules in these cells are also composed of calcium and phosphorus.

Essential revisions:1) The demonstration of MCU independence for granule formation is strong, but is there anything known about what happens to these granules in the absence of the Pi carrier, or when it is mutated?

Deletion of SLC25A3, the Pi carrier, is much more deleterious to cells and organisms than is deletion of MCU. Our understanding is that a total deletion of SLC25A3 in mice is embryonic lethal. An inducible cardiac-specific knockout has been made (Kwong et al., Cell Death Diff.21, 1209-1217 (2014)), but we have not yet been able to grow beating cardiac cells on grids, complicating a comparison of wild-type and PiC knockouts in this cell type.

We did investigate a SLC25A3 knockdown in primary fibroblasts. The knockdown was strong as assessed by RT-PCR, and a SLC25A3 knockdown cell culture grown on glass slides showed slightly increased red fluorescence (hyperpolarization) after JC-1 staining compared to mock-transfected cells. Though we performed a survey of a SLC25A3 knockdown cell culture using CSTET, we cannot guarantee that all cells imaged had lower levels of the carrier, nor did we know the amount of residual carrier remaining even for cells in which SLC25A3 transcript was depleted. We observed elongated and apparently fusing mitochondria containing granules in the SLC25A3 knockdown cultures, but control cell cultures subjected to the knockout procedure with a random oligo did not look appreciably different. Due to the lack of insight provided by these experiments, we did not include them in the manuscript.

2) Are these stores also retrieved in S. cerevisiae that lacks MCU and where the ER is not a Ca^2+^ store?

We did not examine yeast because they are not amenable to CSTET without cryo-FIB milling, which is outside our current capabilities. However, others do not observe granules in yeast mitochondria inside cells (E Villa, personal communication, August, 2017).

3) The demonstration of calcium and phosphor-us signal by EDX for the mitochondrion shown in Figure 2 is convincing. But given that the authors also observe increased density for the mitochondrial matrix compared to cytoplasm by CSET and attribute this to ions, the study should show the EDX signal for a mitochondrion that does not possess dense granules. Also, can imaging and spectroscopy be performed directly on amorphous calcium phosphate for comparison?

Because the EDX is performed on fully hydrated samples, there is insufficient signal-to-noise to distinguish soluble mitochondrial matrix contents from cytoplasm. Only when granules were present was a difference over background cytosol observed. To clarify, the following statement has been added to the Results section:

“Due to the high background in fully hydrated cells, we were not able to obtain sufficient EDX signal from mitochondrial regions lacking granules to compare soluble mitochondrial ion content to cytosol using this technique. This limitation was not surprising, considering that the concentration of matrix calcium ions would reach a maximum of only 100 mM at electrochemical equilibrium, and actual steady-state levels are considerably lower (Pozzan et al., 2000), whereas the concentrations of ions in solid deposits are much higher, as shown below.”

At the reviewer’s request, we have performed imaging and spectroscopy on amorphous calcium phosphate for comparison. The thin film of vitreous ice in which the amorphous calcium phosphate was embedded was suitable for cryo-TEM, so we performed the comparison in TEM mode. Results have been added to Figure 2, and the following text has been added to the Results:

“For comparison with granules formed in mitochondria, biomimetic amorphous calcium phosphate particles were prepared in solution as described (Habraken et al., 2013). […] The general features and appearance of the mitochondrial and synthetic granules are similar, as are the EDX spectra (Figure 2).”

4) In reporting observations on granule presence and distribution in normal cells and after treatment with doxorubicin or FCCP, the authors should give the number of mitochondria imaged and the number with/without granules or specific granule variations (e.g. size variation in Figure 3). This is the only way a reader can assess whether the changes shown in the image examples are really of significance.

We have performed the accounting that the reviewers requested and added the information to the Materials and methods section.

5) The importance of the JC-1 staining experiment to the paper overall should be made clear.

The following sentence has been added to clarify:

“This experiment demonstrates that correlative fluorescence/CSTET studies are feasible but also emphasizes that fluorescent sensors may perturb organelle chemistry and morphology, effects that are readily seen in CSTET.”

6) The number of replicates of experiments should be reported directly in the manuscript and not just in the eLife transparent report form.

We have performed the accounting that the reviewers requested and added the information to the Materials and methods section.